# Prevalence of and Impact on the Outcome of Myosteatosis in Patients with Hepatocellular Carcinoma: A Systematic Review and Meta-Analysis

**DOI:** 10.3390/cancers16050952

**Published:** 2024-02-27

**Authors:** Aikaterini Kamiliou, Vasileios Lekakis, George Xynos, Evangelos Cholongitas

**Affiliations:** 1First Department of Internal Medicine, Laiko General Hospital, Medical School of National and Kapodistrian University of Athens, 11527 Athens, Greece; aikkamiliou@gmail.com (A.K.);; 2Academic Department of Gastroenterology, Laiko General Hospital, Medical School of National and Kapodistrian University of Athens, 11527 Athens, Greece; dp4521909@hua.gr

**Keywords:** poor muscle quality, muscle fat infiltration, hepatocellular carcinoma, cirrhosis, end stage liver disease, prognosis, outcome, non-alcoholic fatty liver disease, Child–Pugh, frailty

## Abstract

**Simple Summary:**

Hepatocellular carcinoma (HCC) is recognized as one of the most prevalent malignancies worldwide, presenting a substantial healthcare challenge. Myosteatosis, known as the accumulation of fat in the muscles, has raised an escalating interest among patients with several malignancies. The aim of our systematic review/meta-analysis was to assess the prevalence of myosteatosis in individuals diagnosed with HCC. Our study revealed that myosteatosis is highly prevalent in HCC patients and is associated with more severe underlying liver disease and higher mortality rates. Our findings also suggest that the prevalence of myosteatosis in HCC patients varies depending on the etiology of the liver disease, while variations in myosteatosis prevalence were observed regardless of whether body mass index-based or gender-based criteria were used.

**Abstract:**

Background: Limited data exist on the prevalence of myosteatosis (i.e., excess accumulation of fat in skeletal muscles) in hepatocellular carcinoma (HCC) patients, and no systematic review or meta-analysis has been conducted in this context. Methods: We searched for articles published from inception until November 2023 to assess the prevalence of myosteatosis in patients with HCC. Results: Ten studies with 3316 patients focusing on myosteatosis and HCC were included. The overall prevalence of myosteatosis in HCC patients was 50% [95% Confidence Interval (CI): 35–65%]. Using the body mass index-based criteria (two studies), the prevalence was 34%, while gender-based criteria (eight studies) yielded 54% (*p* = 0.31). In Asian studies (n = 8), the prevalence was 45%, compared to 69% in non-Asian countries (two studies) (*p* = 0.02). For viral-associated HCC (eight studies), the prevalence was 49%, rising to 65% in non-alcoholic fatty liver disease-associated cases (three studies) and 86% in alcoholic liver disease-associated cases (three studies) (*p* < 0.01). The prevalence of myosteatosis was higher in Child–Pugh class C (3 studies, 91%) than in A (7 studies, 73%) or B (6 studies, 50%) (*p* = 0.02), but with no difference between Barcelona Clinic Liver Cancer stage A (3 studies, 66%), B (4 studies, 44%) and C (3 studies, 62%) (*p* = 0.80). Patients with myosteatosis had a significantly higher mortality (six studies) (Relative Risk: 1.35 (95%CI: 1.13–1.62, *p* < 0.01). Conclusion: The prevalence of myosteatosis is high in HCC patients and is associated with more severe liver disease and higher mortality rates.

## 1. Introduction

Hepatocellular carcinoma (HCC) accounts for 70–85% of all cases of primary liver cancer [1,2] and is globally the third leading cause of cancer-related mortality, with a 5-year survival rate of about 18% [1,2].

In recent years, anthropometric parameters associated with skeletal muscle quantity and quality have been proposed as potential prognostic factors in patients with different comorbidities [3]. Myosteatosis is a relatively novel index of muscle composition, defined as the excess accumulation of fat in skeletal muscles (inter- or intramuscularly) leading to disrupted contractility and impaired function [3]. Muscle biopsy is considered the gold standard for the evaluation of adipose tissue infiltration, but in clinical practice, the evaluation of myosteatosis is usually based on computed tomography (CT) or magnetic resonance imaging (MRI), which can assess muscle attenuation at a specific cross-sectional muscle area [3]. However, it should be mentioned that the exact mechanisms implicated in the pathogenesis of mysteatosis have not been elucidated, while no standardized criteria regarding the optimal cutoffs for the diagnosis of myosteatosis have been established [3]. Thus, although most studies have been using cutoffs with gender-based criteria, in other studies the diagnosis of myosteatosis is based on the body mass index (BMI) usually with a cutoff of ≥25 kg/m^2^ [3]. 

Nevertheless, a previous meta-analysis revealed that myosteatosis is an important prognostic factor of adverse outcomes in oncologic patients across multiple cancer types attributable to their decreased physical performance leading to a poor quality of life, and increased frailty, morbidity, and mortality [4]. In fact, it is considered to be that myosteatosis is highly prevalent in patients with malignancy because of systemic inflammation, nutritional deterioration, and metabolic abnormalities during cancer progression [4]. Myosteatosis in patients with HCC has been assessed in limited individual studies that have yielded contradictory results regarding its prevalence. Interestingly, a recent meta-analysis evaluated the impact of myosteatosis on different types of cancers showing that the patients with myosteatosis had a greater mortality risk compared to those without myosteatosis, particularly among those with gynecological, renal, gastric and colon carcinoma, as well as with HCC (the latter was based on only three studies) [4]. Thus, no meta-analysis focusing on HCC has been performed. This analysis would elucidate the frequency of myosteatosis in HCC patients taking into consideration the severity of the underlying liver disease and the stage of HCC, and would evaluate the association between the presence of myosteatosis with different outcomes of HCC patients and the need to improve their management. Thus, our aim was to perform a comprehensive systematic review and meta-analysis to evaluate the prevalence of myosteatosis in patients with HCC overall as well as in specific different subgroups and to evaluate its prognostic impact on patients with HCC. 

## 2. Methods

### 2.1. Data Sources and Searches

The Medline/PubMed, Embase and Cochrane databases were searched for studies published from inception until November 2023 according to the Preferred Reporting Items for Systematic Reviews and Meta-Analyses (PRISMA) and the guidelines of the *Cochrane Handbook for Systematic Reviews of Interventions* to identify all medical literature included under the keywords “myosteatosis” or “muscle quality” or “muscle alterations” AND “hepatocellular carcinoma” OR “primary liver cancer”. In addition, we searched all relevant reviews as well as the major hepatology congresses during the last year to identify further original articles. Finally, there was no evaluation of grey literature, but we checked the reference lists of the included studies to find additional eligible studies. The protocol has not been registered.

### 2.2. Study Selection

Eligibility criteria were defined using the PICO statement: P: adult patients with a confirmed diagnosis of HCC; I: myosteatosis identified through any of the definitions currently in practice (BMI-based; gender-based); C: adult patients diagnosed with HCC and without myosteatosis; O: to determine the prevalence of myosteatosis in HCC patients taking into account the stage of the disease and the severity of the underlying liver disease, as well as to assess the link between myosteatosis and various HCC patient outcomes.

Only studies published in the English language without country restriction were considered eligible if they fulfilled all the following criteria: (1) they were randomized controlled trials or observational cohort studies, (2) they included adult patients (>18 years) with HCC, (3) the definition of myosteatosis was provided, and (4) the prevalence of myosteatosis was reported. In each selected study, only patients with HCC were evaluated whenever this was possible. Two reviewers (AK, GX) performed the literature search for relevant studies to determine the eligibility for further evaluation based on their titles and abstracts. Each study in the list of the preselected papers was assessed by two reviewers (EC, VL) independently to determine whether it fulfilled all the inclusion criteria. The exclusion criteria were case reports and review articles as well as studies including patients <18 years old or patients suffering from non-HCC neoplasms.

### 2.3. Data Extraction and Quality Assessment

Τwo authors (AK, GX) extracted the following data from the finally selected articles: first author, date of publication, country of origin, type of study, sample size, gender, mean or median age, definition of myosteatosis and the method for its evaluation recording the specific cutoffs to define myosteatosis, aetiology of the underlying liver disease [viral, non-alcoholic fatty liver disease (NAFLD), alcoholic liver disease (ALD) or other], severity of liver disease based on Child–Pugh (CP) class (CP A, B or C) and the number of patients with type II diabetes mellitus (T2DM). In addition, the number of patients in each stage of HCC based on the Barcelona Clinic Liver Cancer (BCLC) system and Albumin-Bilirubin (ALBI) grade and the type of anti-cancer therapy (e.g., chemoembolization, hepatic resection, systemic therapy) were also recorded. The same data were extracted from patients with or without myosteatosis, whenever available. Finally, the mortality or survival, complications and recurrence/response rates were also recorded in the total cohorts, as well as the patients with and without myosteatosis whenever available. 

### 2.4. Data Synthesis and Analysis

We used a descriptive approach to summarize study characteristics and outcomes with regard to the presence of myosteastosis. Quantitative variables were expressed as mean values ± standard deviation and/or median values along with the corresponding ranges. The level of significance was set to 0.05, thus, tests with *p*-values less than 0.05 were considered statistically significant. 

The meta-analysis was performed using a generalized linear mixed model (GLMM) [5]. The two-sided confidence intervals for the single proportions of each individual study were computed using the Clopper and Pearson method [6]. The between-study variance component (τ^2)^ was estimated applying the maximum likelihood method, based on marginal distribution [7]. I^2^ was used to measure heterogeneity, and an I^2^ value of 25%, 50% and 75% represented low, moderate, and high degrees of heterogeneity, respectively. Random effects were used for all calculations [8]. The pooled proportions along with the 95% confidence intervals (CI) and the prediction intervals (PI) were calculated [9]. Regarding binary outcomes, the pooled relative risk ratio (RR) and 95% CIs were used to investigate the impact of myosteatosis on the incidence of death and the recurrence/no response using random effects modeling (DerSimonian-Laird Method). The analysis was conducted in R v4.1.2 using meta-packages and metaprop functions [10]. Statistical analyses were performed by VL (M.Sc. in Research methodology in biomedicine, biostatistics and clinical bioinformatics, Laboratory of Biomathematics, University of Thessaly, Medical school, Greece).

## 3. Results

In total, 27 articles were initially identified from the literature search, but only 12 studies fulfilled the inclusion criteria and underwent further evaluation (Appendix A) [11,12,13,14,15,16,17,18,19,20,21,22]. Three studies from a single center in Japan [17,21,22] had overlapping study periods, and therefore only the most recent study [17] was included. Thus, 10 studies [11,12,13,14,15,16,17,18,19,20], that evaluated the prevalence of myosteatosis in HCC patients, fulfilled all inclusion criteria and were included in the final analysis. Four studies were derived from Japan [11,15,17,19] and one each from Thailand [12], Germany [13], China [14], Italy [16], Taiwan [18] and Indonesia [20]. MRI was used for the evaluation of myosteatosis in only one study [11]. Eight of the ten studies had a retrospective design [11,12,13,14,16,17,18,19]. The Newcastle-Ottawa scale (NOS) was used to assess the quality of the included studies [23]. Based on that, the studies had a low risk of bias (NOS scored > 5) (Appendix A).

### 3.1. Characteristics of Patients

In total, 3316 patients with HCC [mean age: 63.4 years, 72.5% (2406/3316) males] were evaluated. In most patients (93.6% or 3105/3316), the diagnosis of myosteatosis was defined using the gender-based definition with different cutoffs between males and females (e.g., <39.3 HU in females and <44.4 HU in males), while in two studies that included 211 patients, myosteatosis was defined as having a muscle/m^2^ radiodensity at the third lumbar vertebra of <41 HU for patients with a dry BMI < 25 kg/m^2^ and <33 HU for those with a ΒΜΙ ≥ 25 kg/m^2^ (i.e., BMI-based definition) [13,18]. According to the available data, chronic viral hepatitis (B or C) was the underlying cause of liver disease in 72.5% (2404/3316) of patients, while 28.9% of the patients (854/2953) had T2DM [11,12,15,16,17,19]. In addition, 92.4% (3065/3316) of patients were from Asia, and among the 3216 patients with available data, 2633 (81.9%), 561 (17.5%) and 22 (0.6%) of them were classified as CP class A, B and C, respectively [11,12,14,15,16,17,18,19,20].

#### 3.1.1. Characteristics of Patients with Myosteatosis

In total, 1972 patients with HCC [mean age: 67.2 years, 67.9% (1004/1477) males] had myosteatosis. According to the available data, chronic viral hepatitis (B or C) was the underlying cause of chronic liver disease in 74.8% (1106/1477) of patients, 91% (n = 1797) patients were from Asia, 41.9% (530/1265) were in BCLC A, while among the 1417 patients with available data, 1106 (78%), 291 (20.5%) and 20 (1.5%) of them were classified as CP class A, B and C, respectively. Finally, complications after therapeutic manipulations for HCC were reported in 15.6% (86/548) of patients [11,12,13,16,19], 71.8% (1158/1611) died [12,15,16,18,19,20] and 41.8% (133/318) had a poor response to or recurrence after anti-HCC therapy [11,12,17,18,19] (Appendix A).

#### 3.1.2. Characteristics of Patients without Myosteatosis

In total, 1344 patients with HCC had no myosteatosis [mean age 59.4 years, 76% (473/622) males]. Chronic viral hepatitis (B or C) was the underlying cause of chronic liver disease in 74.6% (464/622) of patients, 94.3% (1268/1344) of patients were from Asia, 30.2% (107/354) were in BCLC A, while, among the 487 patients with available data, 412 (84.6%), 73 (15%) and 2 (0.4%) were classified as CP class A, B and C, respectively. Finally, complications after therapeutic manipulations for HCC were reported in 12.6% (81/642) of patients [11,12,13,16,19], 62.7% (499/795) died [12,15,16,18,19,20] and 30.3% (179/591) had a poor response to or recurrence after anti-HCC therapy [11,12,17,18,19] (Appendix A).

### 3.2. Prevalence of Myosteatosis in Total and in Specific Subgroups

The overall pooled prevalence of myosteatosis in HCC patients was 50% (95% CI: 35–65%; heterogeneity, *p* < 0.01, primary study range 14–85%) (Figure 1) [11,12,13,14,15,16,17,18,19,20]. Τhe pooled prevalence of myosteatosis was 34% (95% CI: 10–70%; heterogeneity, *p* < 0.01) and 54% (95% CI: 40–68%; heterogeneity, *p* < 0.01) in studies using the BMI-based definition and gender-based definition, respectively (*p* = 0.31) (Figure 2). However, a significant difference in the pooled prevalence of myosteatosis was found between studies from Asia in comparison with non-Asian countries [45% (95% CI: 30–62%; heterogeneity, *p* < 0.01) vs. 69% (95% CI: 57–79%; heterogeneity, *p* < 0.01), respectively, *p* = 0.02] (Figure 3). Finally, no difference in the pooled prevalence of myosteatosis was found between patients (a) with or without T2DM [65% (95% CI: 35–86%; heterogeneity, *p* < 0.01) vs. 69% (95% CI: 52–82%; heterogeneity, *p* < 0.01, respectively, *p* = 0.80] and (b) those who received TACE, compared to those who underwent hepatectomy or various therapeutic manipulations (mixed hepatectomy, surgery and systemic therapy) [58% (95% CI: 31–81%; heterogeneity, *p* < 0.01) vs. 42% (95% CI: 36–49%; heterogeneity, *p* = 0.02) vs. 63% (95% CI: 41–80%; heterogeneity, *p* < 0.01), respectively, *p* = 0.13].

Gender. The pooled prevalence of myosteatosis was similar between men and women [53% (95% CI: 35–71%; heterogeneity, *p* < 0.01) vs. 52% (95% CI: 32–71%; heterogeneity, *p* < 0.01), *p* = 0.92] (Appendix A), regardless of the definition criteria or geographical area. 

Etiology of liver disease. The pooled prevalence of myosteatosis was significantly lower in patients with viral-associated HCC, compared to those with NAFLD-associated and ALD-associated HCC [49% (95% CI: 30–68%;heterogeneity, *p* < 0.01) vs. 65% (95% CI: 51–77%; heterogeneity, *p* = 0.08) vs. 86% (95% CI: 81–90%; heterogeneity, *p* = 0.79), respectively, *p* < 0.01] (Figure 4). Interestingly, the pooled prevalence of myosteatosis between NAFLD- and ALD-associated HCC was also significant (*p* = 0.02). 

Severity of liver disease. Based on the available data, myosteatosis was significantly more frequent in patients with more severe liver disease, since the prevalence of myosteatosis was 50% (95% CI: 30–70%; heterogeneity, *p* < 0.01), 73% (95% CI: 45–89%; heterogeneity, *p* < 0.01) and 91% (95% CI: 70–98%; heterogeneity, *p* = 0.94) in patients at CP class A, B and C, respectively (*p* = 0.02) (Figure 5). 

Myosteatosis in different ALBI and BCLC stages. Based on the available data, myosteatosis was similar between ALBI grade 1, 2 and 3 [67% (95% CI: 47–82%; heterogeneity, *p* = 0.02), 78% (95% CI: 70–84%; heterogeneity, *p* = 0.39) and 63% (95% CI: 9–97%; heterogeneity, *p* = 1.0), respectively (*p* = 0.45)]. In addition, myosteatosis was similar between BCLC stage A, B and C [66% (95% CI: 28–91%; heterogeneity, *p* < 0.01), 44% (95% CI: 9–86%; heterogeneity, *p* < 0.01) and 62% (95% CI: 21–91%; heterogeneity, *p* < 0.01), respectively (*p* = 0.80)] (Figure 6). 

### 3.3. Outcome of Patients with and without Myosteatosis

Patients with myosteatosis had significantly lower survival rates, compared to those without myosteatosis [RR: 1.35 (95% CI: 1.13–1.62, *p* < 0.01] (Figure 7). Although patients with myosteatosis had higher recurrence/no response rates of HCC, compared to those without myosteatosis, this difference was not statistically significant [RR: 1.22 (95% CI: 0.90–1.66, *p* = 0.20] (Figure 8). 

### 3.4. Publication Bias

In order to evaluate the existence of publication bias, a funnel plot asymmetry test and Egger’s test were performed [24] (Appendix A). No substantial asymmetry was revealed, as evidenced by the non-significant Egger’s test for a regression intercept (β0: −4.10; SE: 6.05; t: −0.68; *p* = 0.52). The significant variation in findings across individual studies was addressed by employing the random effect model for all calculations and by conducting subgroup analyses based on specific criteria, including the different myosteatosis definitions, the country where the studies were conducted, the gender of patients as well as the severity and etiology of their liver disease.

## 4. Discussion

Accumulating data indicate that myosteatosis adversely affects outcomes in individuals with colorectal cancer, while its impact has also been investigated in patients with lung and esophageal cancer [25,26,27,28], showing that myosteatosis is an indicator of unfavorable overall and progression-free survival. However, limited knowledge exists regarding myosteatosis and its influence on patients with HCC. To the best of our knowledge, this is the first systematic review/meta-analysis that has evaluated the prevalence of myosteatosis in patients with HCC (overall and in several subgroups), as well as its impact on the outcomes. Based on the current literature, which included 10 relevant studies with 3316 HCC patients, we showed that the pooled prevalence of myosteatosis in patients with HCC was 50% (95% CI: 35–65%, primary study range 14–85%) suggesting the substantial presence of myosteatosis within this specific population (Figure 1). 

Although there is no consensus regarding the criteria and the specific cut-offs used for the diagnosis of myosteatosis, it is considered that the gender-based definition of myosteatosis might be more accurate, compared to the BMI-based definition, since women have a greater amount of body fat than men for the same BMI [29]. Interestingly, the prevalence of myosteatosis was higher in studies utilizing the gender-based criteria, compared to those utilizing the BMI-based criteria (54% vs. 34%), although this difference was not significant (*p* = 0.31), possibly because only two studies used the latter criteria (Figure 2). It is also noteworthy that although Asian descent is associated with a higher body fat percentage compared to Caucasian descent with a similar BMI [29], the pooled prevalence of myosteatosis was significantly higher in studies from non-Asian countries compared to those from Asian countries (69% vs. 45%, *p* = 0.02) (Figure 3). However, this finding was based on only two non-Asian studies, which possibly included patients with more advanced liver disease [13,16]. Nevertheless, this result may indicate that ethnicity is an important variable, which might be considered in the criteria for the diagnosis of myosteatosis.

The literature data have revealed the close association between myosteatosis and insulin resistance and thus, the higher prevalence of myosteatosis in patients with metabolic syndrome and NAFLD [30]. Our meta-analysis confirmed these findings, since myosteatosis was more frequently diagnosed in patients with NAFLD-associated HCC, compared to viral-associated HCC (65% vs. 49%), highlighting the potential impact of underlying liver disease on the presence of myosteatosis. In addition, myosteatosis had the highest prevalence in ALD-associated HCC patients, confirming previous studies showing the relationship between alcohol consumption and the development of myosteatosis [31] (Figure 4). 

In our meta-analysis we found that myosteatosis was significantly more prevalent in patients with CP class B and C than CP class A (73% and 91% vs. 50%, respectively, *p* = 0.02) (Figure 5). This finding is consistent with previous studies in which myosteatosis was more prominent in patients with advanced liver disease possibly due to the presence of hyperammonemia, hyper-endotoxemia and malnutrition [32]. Interestingly, no difference was found in the prevalence of myosteatosis in different stages of HCC based on the ALBI grade and BCLC classification, although these findings were based on a limited number of studies (Figure 6). However, patients who underwent hepatectomy (i.e., were at earlier stages of HCC), compared to those who received TACE (i.e., having more advanced HCC), had a lower prevalence of myosteatosis (42% vs. 58%), but this difference was not significant. Nevertheless, although further studies are needed, it seems that the mechanisms implicated in the pathogenesis of myosteatosis are multifactorial including HCC- as well as cirrhosis- and metabolic-associated parameters.

The exact determinants associated with the outcomes of HCC patients have not been fully elucidated, but body composition seems to represent a newly identifiable prognostic index, which affects the prognosis of HCC patients. In this context, a recently published meta-analysis has shown that sarcopenia, which is characterized by the progressive loss of skeletal muscle mass and strength, was associated with inferior survival and a higher risk for HCC recurrence [33]. Myosteatosis reflects the presence of poor skeletal muscle quality and several meta-analyses have associated its presence with a higher risk of adverse outcomes in different malignancies [27,34,35]. Interestingly, a previous meta-analysis included studies with different cancer types, three of which evaluated the presence of myosteatosis in patients with HCC [4]. Although the authors found that myosteatosis was predictive of a poor outcome in HCC patients (HR 1.88 95% CI 1.40–2.52, *p* < 0.0001), no further data on the characteristics of these patients were provided, no subgroup analysis was performed, while the association between myosteatosis with other outcomes was not assessed. A more recent meta-analysis evaluated the prognostic impact of sarcopenia and myosteatosis on HCC patients treated with TACE [36]: although sarcopenia was associated with a poor outcome, no significant association was demonstrated between the presence of myosteatosis and overall survival (HR: 1.29, 95% CI: 0.74–2.25, *p* = 0.366). However, in this meta-analysis [36], only two studies that examined myosteatosis were included and all HCC patients underwent TACE. Furthermore, similar to the previous meta-analysis [4], several limitations could be mentioned, since no assessment of other outcomes was performed, while further subgroup analyses were not provided.

Myosteatosis has also been evaluated in various other clinical conditions. In a recent study including 20,986 subjects, the presence of myosteatosis was independently associated with a higher risk of T2DM [37], while in another study with 2964 participants, those with T2DM and impaired glucose tolerance had greater intermuscular fat despite having identical levels of subcutaneous thigh fat. In the same study, among the subjects with a BMI < 25 mg/kg^2^, higher rates of intermuscular fat and visceral abdominal fat were linked to higher levels of fasting insulin [38]. The latter finding was confirmed in a recent study, in which the presence of myosteatosis was associated with insulin resistance in patients with T2DM, and this effect was more significant in older T2DM patients [39]. Interestingly, in patients with inflammatory bowel disease, although myosteatosis did not increase overall morbidity and mortality, it was associated with a higher postoperative morbidity following bowel resection [40]. In addition, several studies have revealed the negative impact of myosteatosis on the clinical course and outcome of patients with COVID-19 infection [41,42], indicating the importance of the measurement of body composition as a potential imaging biomarker for predicting outcomes in patients with viral and/or bacterial infections.

To our knowledge, this is the first systematic review/meta-analysis focusing on HCC without any restriction in inclusion criteria regarding HCC characteristics or therapeutic manipulations. In addition, it is the first which has evaluated the association between myosteatosis and survival, as well as complications and response rates in HCC. We found that HCC patients with myosteatosis had worse survival, compared to those without myosteatosis (RR: 1.35 (95% CI: 1.13–1.62, *p* < 0.01)) (Figure 7). Interestingly, patients with myosteatotis had higher recurrence/no response rates of HCC, compared to those without myosteatosis, although this difference was not significant (RR: 1.22 (95% CI: 0.90–1.66, *p* = 0.20)) (Figure 8). The mechanisms involved in the adverse impact of myosteatosis on HCC patients remain unclear, but they might be related with the presence of malnutrition, muscle weakness and frailty, as well as the development of an inflammatory imbalance and/or a dysfunction of the immune system [43,44].

However, it is important to acknowledge some limitations of our meta-analysis. Although the included studies had a relatively high NOS score, many outcomes had high heterogeneity, the total number of studies used was low and each study had a relatively small number of patients, while eight of the ten included studies were retrospective in nature, indicating the presence of a possible selection bias. Furthermore, only two studies were from non-Asian countries which limits the generalizability of our findings on an international scale. Therefore, further studies including diverse populations and considering important variables such as ethnicity to define myosteatosis are needed to confirm our conclusions. In addition, due to the limited available data, we were not able to identify the causes of death (liver-related or not), as well as separately analyze the outcomes based on the type of therapeutic manipulation, the stage of HCC or the severity of the underlying liver disease. 

## 5. Conclusions

The present systematic review/meta-analysis is the first focused on myosteatosis in HCC. We showed that myosteatosis is very prevalent in patients with HCC, it is associated with the severity and the aetiology of the underlying liver disease but not with the stage of HCC. Interestingly, although myosteatosis is not considered a formal determinant of poor prognosis in the international guidelines for HCC, it was found that it had a negative impact on the survival of HCC patients, and it may be an indicator of a lower response to therapeutic manipulations. These findings are important in daily clinical practice for the early detection and incorporation of myosteatosis in the management of HCC patients (including the nutritional support, physical exercise, medication) to improve their outcomes, these implications are potential areas for future research and further studies are warranted to better clarify these issues.

## Figures and Tables

**Figure 1 cancers-16-00952-f001:**
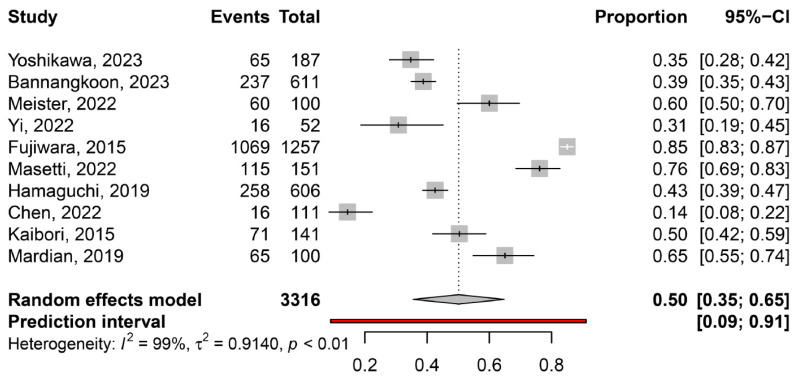
The pooled overall prevalence of myosteatosis in patients with hepatocellular carcinoma (HCC) in the included studies [11,12,13,14,15,16,17,18,19,20].

**Figure 2 cancers-16-00952-f002:**
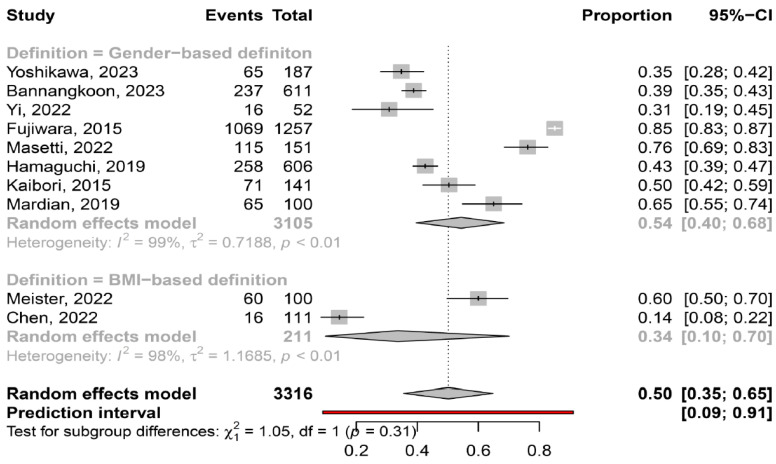
Forest plot of studies comparing the prevalence of myosteatosis according to the definition criteria. BMI: body mass index [11,12,13,14,15,16,17,18,19,20].

**Figure 3 cancers-16-00952-f003:**
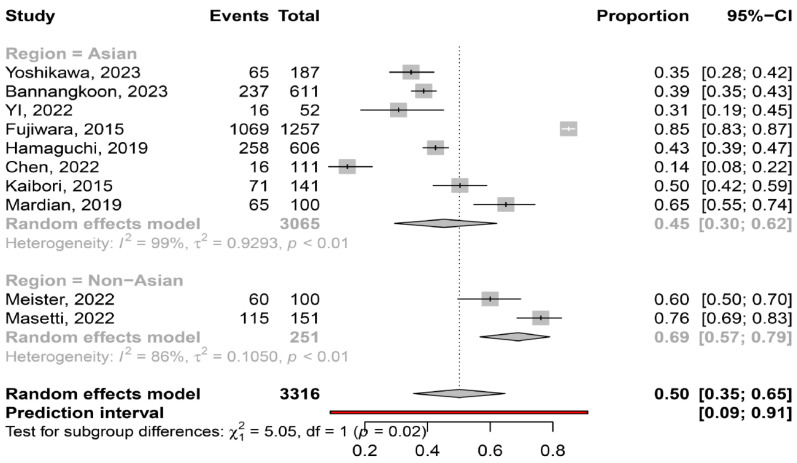
Forest plot of studies comparing the prevalence of myosteatosis according to the region of the studies (Asian vs. non-Asian countries) [11,12,13,14,15,16,17,18,19,20].

**Figure 4 cancers-16-00952-f004:**
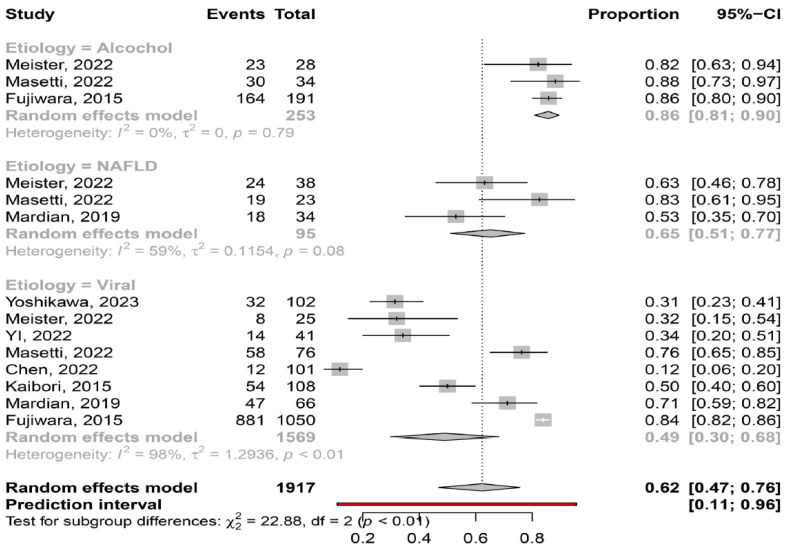
Forest plot of studies comparing the prevalence of myosteatosis according to the aetiology of underlying liver disease (NAFLD: non-alcoholic fatty liver disease) [11,13,14,15,16,18,19,20].

**Figure 5 cancers-16-00952-f005:**
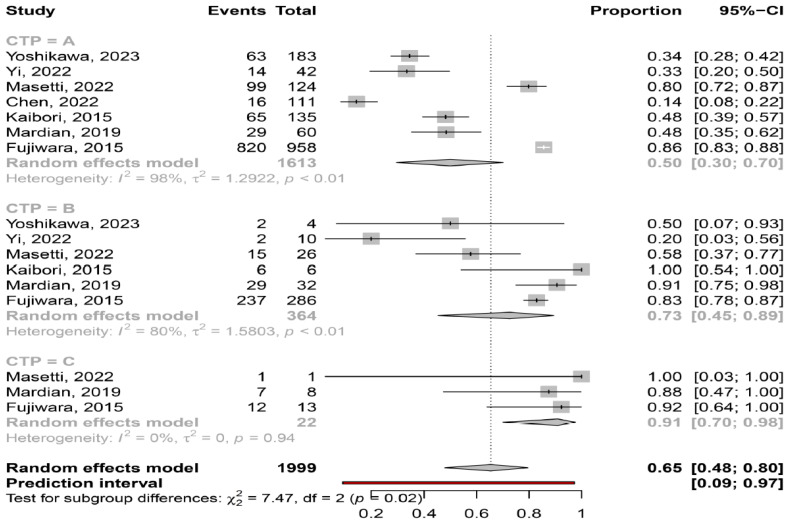
Forest plot of studies comparing the prevalence of myosteatosis according to the Child–Pugh classification [11,14,15,16,18,19,20].

**Figure 6 cancers-16-00952-f006:**
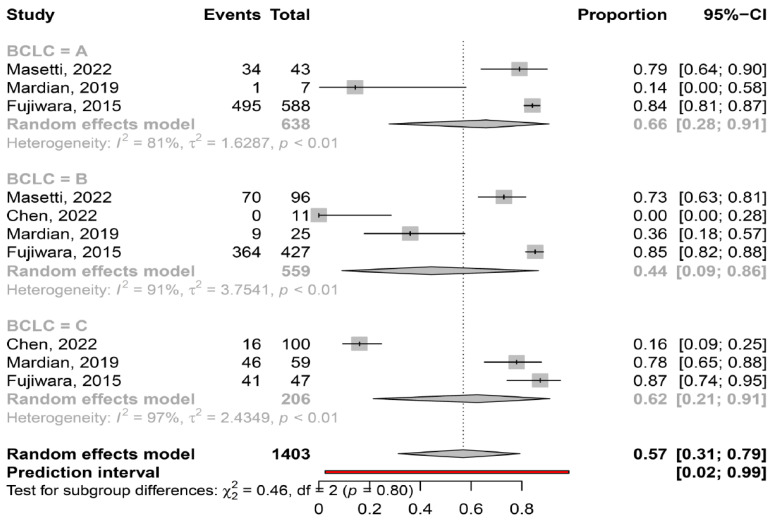
Forest plot of studies comparing the prevalence of myosteatosis according to BCLC classification (BCLC: Barcelona Clinic Liver Cancer) [15,16,18,20].

**Figure 7 cancers-16-00952-f007:**
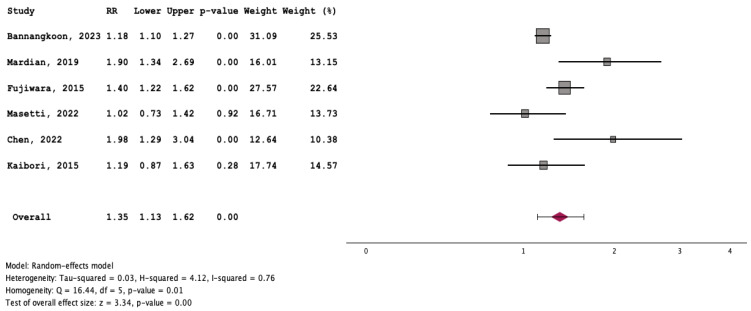
Forest plot for the assessment of an association between myosteatosis and mortality [12,15,16,18,19,20].

**Figure 8 cancers-16-00952-f008:**
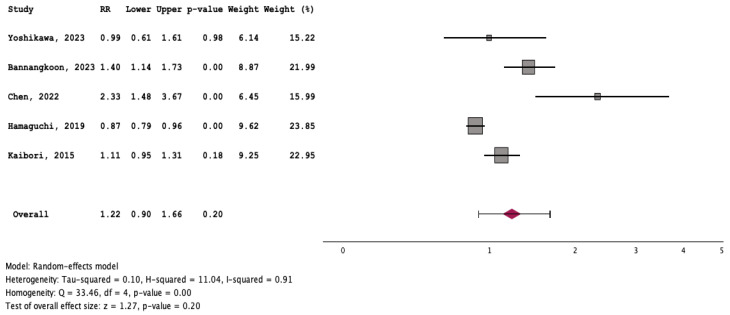
Forest plot for the assessment of an association between myosteatosis and recurrence/no response rates of HCC [11,12,17,18,19].

## Data Availability

No new data were created or analyzed in this study. Data sharing is not applicable to this article.

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
