# Peer review of "Prevalence of and Impact on the Outcome of Myosteatosis in Patients with Hepatocellular Carcinoma: A Systematic Review and Meta-Analysis"

_cancers, 2024, doi:10.3390/cancers16050952_

Round 1
Reviewer 1 Report
Comments and Suggestions for Authors
With pleasure, I read the paper titled “Prevalence and impact on the outcome of myosteatosis in patients with hepatocellular carcinoma: a systematic review and meta-analysis”. The topic is clinically relevant to practice, and of importance to the readers of the Journal CANCERS. Overall, the manuscript reads well and has good flow of ideas, relevant citations, and good summary of data using tables and figures. The main strength includes being the first-ever meta-analysis on the prognostic impact of myosteatosis among patients with HCC. Additional strengths comprise the detailed subgroup analyses, adding robustness to the results. The introduction section was detailed enough to provide the reader with the needful background information, however, some edits are still required. The methods section was detailed too, however, some edits are needed for complete reporting. The results are beautifully presented, however, attention should be paid to PRISMA figure, quality assessment, and Figures 7-8 (See my notes below). The discussion section provided some elaboration and comparison with previously published literature, however, some relevant paragraph could be added to enhance the overall scientific content. The research had some unavoidable limitations, all of which had been explicitly acknowledged, but additional limitations should be elaborated as noted. The conclusion is line with the presented results. All in all, this manuscript is clinically relevant, and this likely to be cited in the future. The manuscript merits publication. However, some changes are required.
ABSTRACT
(a) All abbreviations must be spelled out upon first encounter. Please mention the dates of search from when to when. For each reported outcome, please mention the number of pooled studies. You may want to define myosteatosis in the abstract.
INTRODUCTION
(a) Please briefly describe the gender-based and BMI-based criteria for myosteatosis.
(b) Please clearly highlight the significance of your search. For example, is this the first-ever meta-analysis on the topic?
(c) Please briefly expand further on the impact of myosteatosis in the different cancers by providing some details.
(d) Please conclude the introduction section with some proposed hypotheses.
METHODS
(a) In addition to PRISMA statement, please mention if the guidelines of the Cochrane Handbook for Systematic Reviews of Interventions were followed during the preparation of this research.
(b) Please mention if specific filters (such as year of research, country of publication, or English language) were used during literature screening.
(c) Have you searched the grey literature or the reference lists of the included studies for additional studies that could have been missed?
(d) Could you please kindly provide the rationale for excluding patients with <18 years old?
(e) The inclusion criteria are described well. However, it is recommended to report the inclusion criteria using the standard evidence based PICOS method.
(f) Please indicate when the random-effects and fixed-effects models were used according to the Higgins I2 value.
(g) For testing between-study heterogeneity, besides the Higgins I2 value, did you also consider the p-value (<0.1) of the chi-square Cochran's Q statistic?
(h) Have you examined the risk of publication bias via funnel plots and Egger’s regression test?
(i) Please mention the tool used to evaluate the quality of the included studies.
RESULTS
(a) For Figure 1, I believe the PRISMA flow diagram does not look right. The authors claimed searching three databases and the retrieved number of citations is expected to be higher than just 12. Also, it is expected that some citations are going to overlap between the databases and get excluded. I recommend revisiting this content. This is a major concern.
(b) For transparency, the details of quality assessment according to the different domains of the Newcastle-Ottawa Scale should be reported separately in a supplementary table.
(c) All figures were analyzed using the random-effects model. Hence, I recommend adjusting the methods section accordingly as the fixed-effects model was not used.
(d) Please explain the difference between “95% confidence interval” for the outcome (random-effects model) and the “prediction interval”.
(e) The quality of Figure 8 is not good. Also, I believe labeling for panels A and B is missing. Additionally, I believe there is no need to evaluate the results based on the fixed- and random-effects models. The authors should stick to only the random-effects model as the Higgin’s I2 value was substantial.
(f) For Figure 7 and Figure 8, I recommend the authors to report the findings in the format of relative risk (RR) and 95% confidence interval by directly comparing the rates of mortality (Figure 7) and recurrence/no response (Figure 8) in the myosteatosis group versus no myosteatosis group. This should be similar to a classical forest plot comparing two groups.
(g) You indicated that all studies, but two, were retrospective studies. I am confused; what is the study type of these two exceptional studies?
DISCUSSION
(a) Please briefly discuss why non-Asians had higher prevalence of myosteatosis compared with Asians.
(b) Please discuss in some details the significance of myosteatosis in conditions other than HCC.
(c) Please discuss if myosteatosis is a formal determinant of poor prognosis in HCC in international guidelines.
(d) Please discuss the clinical implications and future directions.
(e) Please acknowledge additional limitations, such as: (a) almost all, if not all, outcomes had high heterogeneity which could be ascribed to differences in study durations, patient characteristics, and diagnosis criteria, and (b) small number of studies and sample size.
OVERALL
(a) The manuscript needs some polishing for minor English language and editing.
(b) For an additional line of validation/accuracy, in view of the large number of the presented figures, please double-check again that results are matched between figures and data in the results section.
Comments on the Quality of English LanguageMinor English editing
Reviewer 2 Report
Comments and Suggestions for Authors
This study sheds light on the significant prevalence of myosteatosis in hepatocellular carcinoma (HCC) patients and its association with more severe liver disease and higher mortality rates. It is important to acknowledge that there have been other papers, including at least 13 papers according to Pubmed, that have also reported on the relationship between myosteatosis and HCC. Furthermore, it is worth noting that there is at least one systematic review/meta-analysis paper (reference #4) available that partially covers the overall survival in myosteatosis and HCC patients, as the authors have cited.
Regarding the claim of contradictory results in the introduction, it is valid to question the specific contradictory findings that the authors referred to. Upon a thorough examination of the results and discussion sections, no contradictory results were highlighted or discussed. This discrepancy should be addressed by the authors to provide clarity and avoid potential confusion.
In the conclusion, the authors mention that their findings have implications for daily clinical practice, such as nutritional support, physical exercise, and medication. However, it is important to note that this study did not investigate or assess the effectiveness of these interventions. Therefore, it is necessary for the authors to clarify that these are potential areas for future research rather than conclusions drawn from their specific study.
Reviewer 3 Report
Comments and Suggestions for Authors
This paper conducts a review on the association between myosteatosis (intramuscular fat accumulation) and HCC (hepatocellular carcinoma). While the abundance of data is appreciated, it appears overwhelming to the readers, and it is not clear which data points are particularly useful. It is requested that the essential data be emphasized in the main text, with less critical information organized as supplementary material or in a supplemental manner. Additionally, it is suggested to include information on the relevance to statisticians in the main text.
Comments on the Quality of English LanguageAlmost English looks good.
Round 2
Reviewer 1 Report
Comments and Suggestions for Authors
The authors did a great job by adequately addressing all the raised comments. The manuscript now reads well, scientifically robust, and intellectually sound. Well-done and the manuscript can be accepted for publication in its current, subject to routine English review during copyediting.
Comments on the Quality of English LanguageThe manuscript can be accepted for publication in its current, subject to routine English review during copyediting.